# Comorbidity Prevalence in Prediabetes and Type 2 Diabetes: A Cross-Sectional Study in a Predominantly Hispanic U.S.–Mexico Border Population

**DOI:** 10.3390/ijerph22050673

**Published:** 2025-04-24

**Authors:** Ricardo X. Noriega, Juan J. Nañez, Emily F. Hartmann, John D. Beard, Chantel D. Sloan-Aagard, Evan L. Thacker

**Affiliations:** 1Department of Public Health, Brigham Young University, Provo, UT 84602, USA; ricardox@byu.edu (R.X.N.); john_beard@byu.edu (J.D.B.); chantel.sloan@byu.edu (C.D.S.-A.); 2Paso del Norte Health Information Exchange, El Paso, TX 79912, USA; jnanez@phixnetwork.org (J.J.N.); ehartmann@phixnetwork.org (E.F.H.)

**Keywords:** prediabetes, type 2 diabetes, comorbidity, prevalence, Hispanic

## Abstract

Type 2 diabetes and prediabetes are associated with a higher risk of several health conditions. We conducted a cross-sectional study to compare the prevalence of comorbidities among 88,724 adults with prediabetes and 12,071 adults with type 2 diabetes in El Paso, Texas, using data from the Paso del Norte Health Information Exchange (PHIX) from 1 January 2021, to 31 January 2023. We estimated prevalence ratios (aPR) adjusted for age decade, gender, and Hispanic ethnicity. Individuals with prediabetes, compared to type 2 diabetes, had lower adjusted prevalence of circulatory (59.1% vs. 80.4%; aPR = 0.82 [95% CI: 0.81–0.84]), genitourinary (44.9% vs. 50.5%; aPR = 0.97 [0.96–0.99]), respiratory (32.0% vs. 35.7%; aPR = 0.94 [0.92–0.97]), neurological (27.4% vs. 32.8%; aPR = 0.91 [0.88–0.94]), blood (21.2% vs. 30.5%; aPR = 0.77 [0.75–0.80]), mental (19.5% vs. 26.1%; aPR = 0.72 [0.69–0.75]), infectious (12.8% vs. 21.5%; aPR = 0.63 [0.60–0.66]), skin (12.2% vs. 14.8%; aPR = 0.82 [0.78–0.86]), and COVID-19 (10.2% vs. 11.9%; aPR = 0.86 [0.81–0.91]) diseases/conditions. Adjusted prevalence was higher among those with prediabetes for musculoskeletal (53.8% vs. 47.0%; aPR = 1.19 [1.17, 1.21]), ear (18.4% vs. 12.9%; aPR = 1.54 [1.47–1.60]), eye (11.1% vs. 7.8%; aPR = 1.52 [1.43, 1.61]), digestive (44.0% vs. 44.0%; aPR = 1.02 [1.00–1.05]), and neoplastic (14.4% vs. 14.5%; aPR = 1.12 [1.06–1.17]) diseases/conditions. People with prediabetes in El Paso, Texas, had a lower prevalence of most comorbidities than those with type 2 diabetes, suggesting that preventing prediabetes from progressing to type 2 diabetes could have a beneficial impact on comorbid disease burden.

## 1. Introduction

Diabetes and prediabetes present challenges to public health. In 2021, diabetes was the eighth leading cause of death in the U.S. [1]. Approximately 38.4 million people in the U.S. have diabetes, and 97.6 million adults have prediabetes, with 80% unaware of their condition [2,3]. Diabetes and prediabetes can result in serious health complications, impacting overall well-being [4,5,6,7,8,9]. The costs associated with diabetes in the U.S. increased from USD 227 billion in 2012 to USD 307 billion in 2022, imposing a burden on healthcare systems and society [1,10]. The importance of prediabetes management may not be adequately emphasized in healthcare or may be underappreciated by patients, leading to insufficient prevention and intervention efforts. For example, as shown in the National Health Interview Survey, only 5% of individuals with prediabetes recalled being told by their physician to participate in a diabetes prevention program [11]. Therefore, it is imperative to raise awareness and enhance prevention and management strategies to mitigate the health and economic consequences of diabetes.

The higher prevalence of comorbidities among persons with type 2 diabetes than those who are normoglycemic is well documented [8,10,12,13,14]. Some studies have explored the difference in the risk of comorbidities between normoglycemic individuals and those with prediabetes [4,5,6,7,8,9,11,15,16,17]. However, only one large U.S. study has compared the prevalence of comorbidities between prediabetes and type 2 diabetes, finding that people with prediabetes had lower unadjusted self-reported prevalence of cardiovascular diseases, arthritis, kidney diseases, cancer, and chronic obstructive pulmonary disease, higher unadjusted self-reported prevalence of depression, and no difference in unadjusted self-reported prevalence of asthma [18]. The above study used data from the Behavioral Risk Factor Surveillance System (BRFSS), which included self-reported diagnoses primarily from Non-Hispanic White participants [18].

In the present study, we prioritized a predominantly Hispanic community, and we ascertained type 2 diabetes, prediabetes, and comorbidities via electronic health records rather than self-report, thereby addressing two important limitations of the aforementioned work conducted in the BRFSS. We aimed to compare the prevalence of comorbid conditions, defined by International Classification of Diseases, Tenth Revision (ICD-10) code groups [19], between individuals with prediabetes and type 2 diabetes in El Paso, Texas. This region is geographically isolated, has a large Hispanic population (82.9%), and serves patients across the US–Mexico border. The prevalence of diabetes in El Paso is thought to be 15%, higher than the national average [20].

## 2. Materials and Methods

In this cross-sectional study, we analyzed data from 88,724 individuals diagnosed with type 2 diabetes and 12,071 individuals with prediabetes living in El Paso, Texas. Paso del Norte Health Information Exchange (PHIX), a 501(c)(3) non-profit organization serving hospitals, physicians, and patients in west Texas and southern New Mexico, provided the data. PHIX’s data include information from eight hospitals, three micro-hospitals, and 35 outpatient healthcare organizations between 1 January 2021 and 31 January 2023.

Our study included individuals aged 18 and older who had received diagnoses assigned during healthcare encounters corresponding to ICD-10 codes for prediabetes (R73.03) or type 2 diabetes mellitus (E11). We classified individuals who had codes for both prediabetes and type 2 diabetes as having type 2 diabetes. Therefore, the type 2 diabetes group and the prediabetes group in our study consisted of unique individuals with no overlap. Demographic data collected included age, gender, Hispanic ethnicity, and race based on U.S. Census categories. We identified individuals as having a comorbidity if they had one or more diagnoses within the following ICD-10 categories: infectious and parasitic diseases (A00–B99), neoplasms (C00–D49), blood and immune disorders (D50–D89), mental and neurodevelopmental health disorders (F01–F99), nervous system diseases (G00–G99), eye diseases (H00–H59), ear and mastoid diseases (H60–H95), circulatory system diseases (I00–I99), respiratory diseases (J00–J99), digestive diseases (K00–K95), skin diseases (L00–L99), musculoskeletal and connective tissue diseases (M00–M99), genitourinary system diseases (N00–N99), and COVID-19 or post-COVID-19 conditions (U00–U85). This classification enabled a comprehensive analysis of the comorbidity burden in the patient sample.

We examined descriptive statistics overall and group differences between people with type 2 diabetes and prediabetes for age decade (≤29, 30–39, 40–49, 50–59, 60–69, 70–79, 80–89, ≥90), gender (male, female), race (White, African American or Black, Pacific Islander, Asian, American Indian/Alaska Native, or Other), and ethnicity (Hispanic or non-Hispanic). To compare group differences in comorbidity prevalence, we estimated the adjusted prevalence difference (aPD) and adjusted prevalence ratio (aPR) for each comorbidity among people with type 2 diabetes and prediabetes using prediabetes as the reference group. We used linear risk regression models to estimate the aPD and log binomial regression models to estimate the aPR and their 95% confidence intervals (CIs). A *p*-value of less than 0.05 was considered statistically significant. Based on existing peer-reviewed research evidence, age, gender, race, and ethnicity influence both our dependent and independent variables [21,22,23]. Therefore, we adjusted for potential confounding by age decade, gender, and Hispanic ethnicity. We did not adjust for race because the El Paso population is predominantly White Hispanic, a large percentage of participants had “unknown” race recorded, and a very small percentage of participants reported non-White race.

For age adjustment, we compared three versions of age and assessed model fit using the Akaike Information Criterion (AIC) [24]. The first model included age decade as a continuous variable (using the values 20, 30, 40, 50, 60, 70, 80, and 90) to capture a linear relationship with the outcome. In the second model, age decade was centered on the median and squared to explore potential nonlinear relationships. In the last model, age decade was analyzed as a categorical variable with eight categories. For each comorbidity, we used the best-fitting age-adjusted model. The categorical age-adjusted model had the best fit for all comorbidity analyses except the PR for musculoskeletal diseases and the PD for circulatory, eye, and musculoskeletal diseases, for which the centered and squared age-adjusted model had the best fit, and the PD for digestive diseases, for which the continuous age-adjusted model had the best fit.

## 3. Results

Most participants were females (55%) and Hispanic (69%). The distributions of age decade, gender, and racial/ethnic composition were similar in the prediabetes and type 2 diabetes groups (Table 1).

In Figure 1, we contrasted the unadjusted prevalence of comorbid disease categories between individuals with prediabetes and those with type 2 diabetes. Individuals within the type 2 diabetes group exhibited a marked prevalence of circulatory system diseases (80.4%) and genitourinary disorders (50.5%). Although musculoskeletal disorders were common in both groups, their prevalence was slightly higher in the prediabetes group (53.8%) than in the type 2 diabetes group (47.0%). Digestive, respiratory, nervous system, and blood diseases were relatively common across both conditions, with a higher prevalence observed in the type 2 diabetes group. In contrast, skin, neoplasm, ear, COVID-19, and eye diseases were less frequent among the studied groups. The prediabetes group exhibited a comparatively lower prevalence of most comorbid conditions than the type 2 diabetes group.

Table 2 contains the aPD and aPR comparing individuals with prediabetes to those with type 2 diabetes. Adjusted PRs exhibited statistically significantly lower prevalences of several disease categories among the prediabetes group, including infectious and parasitic diseases, diseases of the blood, mental and behavioral illnesses, diseases of the nervous system, diseases of the respiratory system, diseases of the circulatory system, diseases of the skin and subcutaneous tissue, diseases of the genitourinary system, and COVID-19-related conditions. The adjusted PR for circulatory system diseases was 0.82 (95% CI: 0.81, 0.84), which indicates an 18% lower prevalence in the prediabetes group. Furthermore, the adjusted PD was −14.6 per 100 individuals (95% CI: −15.7, −13.7), indicating 15 fewer cases of circulatory diseases per 100 people in the prediabetes population. Conversely, higher adjusted prevalences of conditions affecting the eyes, ears, and musculoskeletal system and neoplasms were observed in individuals with prediabetes.

## 4. Discussion

In our study, individuals with prediabetes had lower prevalences of various comorbidities than those with type 2 diabetes. Circulatory, genitourinary, musculoskeletal, digestive, and respiratory system diseases were the top five prevalent conditions in both groups, with the prevalence of circulatory diseases reaching 80% in the type 2 diabetes group and 59% in the prediabetes group. We also found that the largest prevalence differences between these groups were in circulatory, mental, blood, and infectious diseases.

Data from NHANES 2011–2014 underscore the substantial burden of cardiovascular risk factors among individuals with prediabetes, including high prevalences of hypertension (36.6%), dyslipidemia (51.2%), albuminuria (7.7%), and reduced estimated glomerular filtration rate (4.6%) [11]. In a cross-sectional study using BRFSS 2011–2015 data that included 63,567 individuals with prediabetes and 215,441 individuals with type 2 diabetes, people with prediabetes had lower unadjusted self-reported prevalence of cardiovascular diseases (PD: −9.9 per 100), arthritis (PD: −5.7 per 100), kidney diseases (PD: −5.1 per 100), cancer (PD: −2.0 per 100), and chronic obstructive pulmonary disease (PD: −1.4 per 100); while those with prediabetes had higher unadjusted self-reported prevalence of depression (PD: +1.4 per 100) and no difference in unadjusted self-reported prevalence of asthma (PD: +0.1 per 100) [18]. In contrast with those BRFSS findings, we observed in our study that those with prediabetes had 14.6 fewer cases of circulatory disease, 1.3 fewer cases of genitourinary disease, 7.2 fewer cases of mental illness, and 2.1 fewer cases of respiratory disease per 100 but 8.5 more cases of musculoskeletal disease and 1.7 more cases of cancer per 100, adjusted for demographics. Therefore, our findings may differ from those of previous studies because of between-study differences in the characteristics of the target population (predominantly Hispanic versus predominantly non-Hispanic), methods for detecting comorbidities (documented in the medical records versus self-reported), and decisions about covariate adjustment. We also assessed a larger set of comorbidity types than the BRFSS study, which enriched the knowledge base.

Our findings and those of the BRFSS and NHANES emphasize that circulatory diseases represent a major health burden in the population with type 2 diabetes and suggest an opportunity to reduce the emergence of that burden among individuals with prediabetes through the prevention of progression to type 2 diabetes. Individuals with prediabetes face a relative risk of progressing to type 2 diabetes that varies from 4 to 12, depending on the prediabetes definition used [11]. Extensive research shows that prediabetes is a reversible condition, and interventions focused on diet and exercise have proven effective in reducing the occurrence of type 2 diabetes [11,25,26,27,28,29,30,31,32,33]. For instance, in a six-year follow-up study, individuals with prediabetes who engaged in exercise interventions experienced a decrease in type 2 diabetes development [28]. Notable examples include the Diabetes Prevention Program (DPP) and the Diabetes Prevention Study (DPS), which are both large-scale studies that have yielded positive outcomes through lifestyle interventions. The DPP study, for instance, documented a 58% decrease in type 2 diabetes cases after three years of intensive lifestyle interventions, which involved losing 7% of body weight, reducing total caloric intake by 25%, and engaging in 150 min of physical activity weekly. Participants who lost weight and met the recommended physical activity requirements had a risk reduction that exceeded 90%. These findings strongly suggest that emphasizing lifestyle interventions can effectively prevent type 2 diabetes among individuals with prediabetes [28,31,32]. Our finding of a substantially higher burden of circulatory diseases in type 2 diabetes compared with diabetes, even after adjusting for demographic differences, strongly suggests that preventing progression to type 2 diabetes would reduce the occurrence of circulatory diseases.

Other studies have compared people with type 2 diabetes or prediabetes with normoglycemia and revealed more about disease progression or disease burden [4,5,11,17,18]. One study documented an augmented risk of cardiovascular diseases and all causes of mortality among individuals who had prediabetes compared to individuals with normoglycemia. Identifying those at risk of type 2 diabetes through screening and effectively managing those with prediabetes may help prevent cardiovascular diseases, which are the leading cause of death in the U.S. Another study also compared individuals with prediabetes and individuals with normoglycemia. The absolute risk differences reported for various adverse health outcomes, including all-cause mortality, composite cardiovascular disease, coronary heart disease, and stroke, underscore the elevated risks faced by individuals with prediabetes compared to those with normoglycemia [4]. The findings of Zhang et al. show the importance of preventive measures in managing prediabetes and its associated risks. While preventing progression from prediabetes to type 2 diabetes is important, the data suggest that intervening even earlier, at the normoglycemic stage, may provide even greater health benefits.

Our findings have implications for the El Paso, Texas, community. Our results provide a better understanding of the profile of comorbid health conditions among people with prediabetes and type 2 diabetes in the predominantly Hispanic population of El Paso, Texas. Our findings indicate a need for improved discussion on prediabetes management and the implementation of comprehensive, evidence-based health programs in this population. The engagement of a coalition such as the Diabetes Alliance in El Paso, which involves healthcare professionals, researchers, community leaders, and policymakers, exemplifies the collaborative efforts needed to translate these findings into action.

Our findings also have implications for other communities. Collaboration between health information exchanges and clinical care providers could lead to more personalized patient education and empowerment and create a motivating environment for individuals to engage in proactive health behaviors to reduce their risk of developing type 2 diabetes and its associated comorbidities. Health information exchanges such as PHIX could be implemented in additional communities to provide a secure source of healthcare data for epidemiological research and collaborative work. Health information exchanges allow for the implementation of a sharing system for data across health care providers, researchers, and public health agencies to support the complete management of the population and the creation of specific strategies aimed at improving the population’s health. Therefore, the identified functions of health information exchanges can help break the barriers between different disciplines and organizations, gather and analyze information, and, finally, translate findings into interventions.

Our study has methodological strengths. This research offers insights into the health burden of prediabetes and type 2 diabetes in a predominantly Hispanic population. The use of data from PHIX provided a representative sample from the El Paso population, and the large sample size allowed us to make precise estimates of the differences in comorbidity prevalence between the studied groups. To ensure a meaningful analysis, we selected fourteen well-established ICD-10 comorbidity categories, which will enhance the comparability of our findings with future studies and improve the internal validity of our results. Furthermore, we examined both prevalence differences and prevalence ratios to consider the absolute and relative magnitudes of associations between prediabetes, type 2 diabetes, and various comorbidities. Linear risk and log-binomial regression models were used to adjust for confounding demographic factors effectively. The dataset had minimal missing or unknown values for demographics, except for race, which was not adjusted for due to the excessive representation of the categories “unknown” and “other” in the variable. As the population in El Paso is predominantly Hispanic (81.6%), with most identifying as White Hispanics, we considered Hispanic ethnicity to be a more important group identifier than race category.

Our study also has methodological limitations. First, the sample was exclusively from El Paso, Texas, which, while ideal for our research purposes, may not be generalizable to other regions or healthcare systems with more demographic diversity. Second, while using ICD-10 diagnosis codes provides a strength in comparison to prior studies, it also carries the risk of misclassification due to coding inaccuracies. We identified participants as having diabetes or prediabetes based solely on ICD-10 codes assigned during healthcare encounters. We did not directly examine the HbA1c levels of our participants to confirm their diagnoses. Furthermore, the use of broad code categories may have limited the ability to detect or fully explore important associations between prediabetes, type 2 diabetes, and specific diseases within those categories. Third, some associations may have been influenced by unmeasured confounding factors, such as cigarette smoking, physical activity, body mass index (BMI), sleep, cholesterol levels, blood pressure, or other cardiovascular and non-cardiovascular factors. Additionally, the lack of adjustment for clinical variables such as BMI or A1C levels may have led to residual confounding, potentially influencing the magnitude or direction of the observed associations.

## 5. Conclusions

People with prediabetes in El Paso generally had a lower prevalence of comorbid health conditions compared to those with type 2 diabetes. Our estimates of the prevalence of comorbidities in prediabetes and type 2 diabetes may generalize to other communities in the U.S. that have predominantly Hispanic populations. Taking action to prevent prediabetes from progressing to diabetes could have a beneficial impact on the comorbid disease burden. Peer-reviewed research evidence suggests that population-based lifestyle intervention programs are effective in influencing the disease course to prevent or delay the onset of adverse outcomes. Future investigations that are longitudinal, tracking individuals without diabetes through their progression to incident prediabetes and incident type 2 diabetes, with accurate ascertainment of time of onset of comorbidities and finer diagnostic code groups across a wider spectrum of comorbidity diagnoses, may help further elucidate the causal relationships between prediabetes and type 2 diabetes and the risk of comorbidities.

## Figures and Tables

**Figure 1 ijerph-22-00673-f001:**
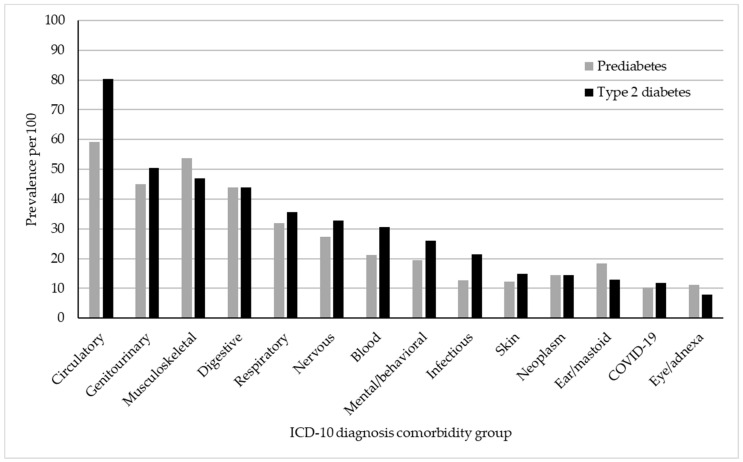
Unadjusted prevalence of ICD-10 diagnosis comorbidity cases per 100 people in the prediabetes and type 2 diabetes samples.

**Table 1 ijerph-22-00673-t001:** Demographic characteristics of participants with prediabetes and type 2 diabetes.

Characteristics	No. (Column %)	
	Diagnosis of Prediabetes (*n* = 12,071)	Diagnosis of Type 2 Diabetes (*n* = 88,724)
**Gender**		
Male	4415 (36.6)	40,176 (45.3)
Female	7618 (63.1)	48,196 (54.3)
Unknown	38 (0.3)	352 (0.4)
**Age decade**		
<29	652 (5.4)	1986 (2.2)
30–39	1031 (8.5)	4442 (5.0)
40–49	2036 (16.9)	9727 (11.0)
40–59	3145 (26.1)	17,468 (19.7)
60–69	3066 (24.4)	24,510 (27.6)
70–79	1446 (12.0)	18,961 (21.4)
80–89	576 (4.8)	9472 (10.7)
>90	119 (1.0)	2158 (2.4)
**Race**		
African American or Black	16 (0.1)	199 (0.2)
Asian	15 (01)	89 (0.1)
American Indian/Alaska Native	3 (0.02)	58 (0.1)
Pacific Islander	40 (0.3)	276 (0.3)
White	7705 (63.8)	57,240 (64.5)
Other	3213 (26.6)	11,005 (12.4)
Unknown	1079 (8.9)	19,857 (22.4)
**Ethnicity**		
Hispanic	8482 (70.3)	61,498 (69.3)
Non-Hispanic	3589 (29.7)	27,226 (30.7)

**Table 2 ijerph-22-00673-t002:** Prevalence differences and prevalence ratios of ICD-10 diagnosis comorbidity groups for prediabetes versus type 2 diabetes.

	Prevalence per 100		Prevalence Difference per 100 (95% CI)		Prevalence Ratio (95% CI)	
	Prediabetes	Type 2 Diabetes	Unadjusted Model	Adjusted Model (Age, Gender, and Ethnicity)	Unadjusted Model	Adjusted Model (Age, Gender, and Ethnicity)
Diseases of the circulatory system	59.1	80.4	−21.3 *** (−22.3, −20.4)	−14.6 *** (−15.7, −13.7)	0.73 *** (0.72, 0.75)	0.82 *** (0.81, 0.84)
Diseases of the genitourinary system	44.9	50.5	−5.6 *** (−6.6, −4.7)	−1.3 ** (−2.3, −0.4)	0.89 *** (0.87, 0.91)	0.97 * (0.96, 0.99)
Diseases of the musculoskeletal and connective tissue	53.8	47.0	6.8 *** (5.8, 7.7)	8.5 *** (7.5, 9.4)	1.14 *** (1.12, 1.16)	1.19 *** (1.17, 1.21)
Diseases of the digestive system	44.0	44.0	−0.01 (−1.0, −0.9)	1.1 * (0.1, 2.0)	1.00 (0.99, 1.02)	1.02 * (1.00, 1.05)
Diseases of the respiratory system	32.0	35.7	−3.7 *** (−4.6, −2.8)	−2.1 *** (−3.0, −1.2)	0.90 *** (0.87, 0.92)	0.94 *** (0.92, 0.97)
Diseases of the nervous system	27.4	32.8	−5.4 *** (−6.3, −4.6)	−3.2 *** (−4.1, −2.4)	0.83 *** (0.81, 0.86)	0.91 *** (0.88, 0.94)
Diseases of the blood	21.2	30.5	−9.28 *** (−10.1, −8.5)	−6.7 *** (−7.4, −5.9)	0.70 *** (0.67, 0.72)	0.77 *** (0.75, 0.80)
Mental and behavioral illnesses	19.5	26.1	−6.6 *** (−7.4, −5.9)	−7.2 *** (−8.0, −6.5)	0.75 *** (0.72, 0.78)	0.72 *** (0.69, 0.75)
Infectious and parasitic diseases	12.8	21.5	−8.6 *** (−9.3, −8.0)	−7.5 *** (−8.2, −6.9)	0.60 *** (0.57, 0.63)	0.63 *** (0.60, 0.66)
Diseases of the skin and subcutaneous tissue	12.2	14.8	−2.6 *** (−3.2, −2.0)	−2.5 *** (−3.1, −1.8)	0.82 *** (0.78, 0.87)	0.82 *** (0.78, 0.86)
Neoplasms	14.4	14.5	−0.1 (−0.8, 0.6)	1.7 (1.0, 2.3)	0.99 (0.95, 1.04)	1.12 *** (1.06, 1.17)
Diseases of the ear and mastoid process	18.4	12.9	5.4 *** (4.7, 6.2)	6.1 *** (5.3, 6.8)	1.42 *** (1.36, 1.48)	1.54 *** (1.47, 1.60)
COVID-19 and post-COVID-19	10.2	11.9	−1.7 *** (−2.3, 1.1)	−1.6 ** (−2.2, −1.0)	0.86 *** (0.81, 91)	0.86 *** (0.81, 0.91)
Diseases of the eye and adnexa	11.1	7.8	3.3 *** (2.7, 3.9)	3.5 *** (2.9, 4.1)	1.42 *** (1.34, 1.50)	1.52 *** (1.43, 1.61)

* *p* < 0.05, ** *p* < 0.01, and *** *p* < 0.001. Abbreviations: CI, confidence interval; ICD-10, International Classification of Diseases, 10th Revision.

## Data Availability

The datasets generated during and/or analyzed in the current study are not directly available due to confidentiality.

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
