# Peer review of "Comorbidity Prevalence in Prediabetes and Type 2 Diabetes: A Cross-Sectional Study in a Predominantly Hispanic U.S.–Mexico Border Population"

_ijerph, 2025, doi:10.3390/ijerph22050673_

Round 1
Reviewer 1 Report
Comments and Suggestions for Authors
Thank you for the opportunity to read this interesting research. It is relevant, overall well written, and the findings inspirational in the context of motivating people with prediabetes to decrease progression to T2DM.
There are a few things things that should be addressed prior to publication
The abstract should specify type 2 diabetes (I'd say mellitus should also be included but it appears that is being left off more and more)
line 36 - chech grammar
line 38 - what "people" - this should be clarified - individuals with pre diabetes? healthcare professionals? and what kind of study? How robust was this evidence?
lines 46-48 - check grammar
In the methods section specifically what method was used to classify people as having prediabetes should be included, as well as the actual HbA1c levels, as the level used to define prediabetes varies from country to country - for example, greater or equal to 42 mmom/L in the UK, and I think 39 mol/L in the USA? If not available then this needs to be an explicityly stated limitation
line 110 - check grammar
line 145 - check grammar
lines 197-200 - the implications of this study are overstated - please modify
Line 204-208 - "there is nothing in this research study that provides evidence to state that "this collaboration will lead" - please modify
Line 251 - and other places where the term "literature" is used - it is the research evidence or at least the research literature - literature alone has a much more broad meaning
I'd suggest adding a sentence or two on what further reserach is actually needed to more definitely prove causation
Reviewer 2 Report
Comments and Suggestions for Authors
[1]. Ln 49-52. Better move this at the very end of the introduction.
[2]. In the study groups, subjects with prediabetes are proportionally more numerous up to age 59, whereas subjects with DM are proportionally more numerous over age 60. The authors could explain/clarify more how they adjusted for age, given that age (and probably duration of the premorbid condition, i.e. prediabetes) also plays a role in the appearance of CVD etc [https://diabetesjournals.org/care/article/41/4/731/36880/Duration-of-Diabetes-and-Prediabetes-During]
Reviewer 3 Report
Comments and Suggestions for Authors
In this manuscript, the authors present prevalent ratio and analysis for population cohorts with diabetes and pre-diabetes. While, in principle, the manuscript is interesting and describes an important topic of research, it is not suitable for publication in its current form.
- How was it ensured that the population cohorts in the two data sets are unique? How would the model behave if input cohorts have overlap (and include patients with co-occurring conditions)?
- Minimal details have been provided regarding the quantitative analysis and the specific results obtained.
- Figure and Table captions are non-existent. This needs to be overhauled.
- Figure 1 needs to incorporate variation in the data set. As it is currently presented, it does not shed any light on the statistical significance of the results.
- I am confused by the caption of Table 2. What does this mean?
- Overall, the analysis presented in the manuscript is very simplistic and minimal. At a minimum, the authors need to quantitate the statistical rigor of the results and dependent of the results on the input data sets. Additionally, can the authors perform quantitative clustering of population analysis to demonstrate any overlap and redundancy in the input data sets that could artificially skew the output?
- There is minimal discussion about the broader applicability of the conclusions presented in the manuscript outside of this study. As such, there does not appear to be a significant mapping to a general population cohort that could be inferred from this manuscript.
Round 2
Reviewer 3 Report
Comments and Suggestions for Authors
The revised manuscript addresses the pertinent questions raised by this reviewer. Additionally, the revisions made regarding other reviewer's comments appear to be comprehensive as well.